# Measuring and Improving the Use of Graph Information in Graph Neural Networks

**Yifan Hou**[1]**, Jian Zhang**[1]**, James Cheng**[1]**, Kaili Ma**[1]**, Richard T. B. Ma**[2]**,**
**Hongzhi Chen**[1]**, Ming-Chang Yang**[1]
[1]Department of Computer Science and Engineering, The Chinese University of Hong Kong
[2]School of Computing, National University of Singapore
{yfhou,jzhang,jcheng,klma}@cse.cuhk.edu.hk    tbma@comp.nus.edu.sg
{hzchen,mcyang}@cse.cuhk.edu.hk

## Abstract

Graph neural networks (GNNs) have been widely used for representation learning on graph data. However, there is limited understanding on how much performance GNNs actually gain from graph data. This paper introduces a context-surrounding GNN framework and proposes two smoothness metrics to measure the quantity and quality of information obtained from graph data. A new GNN model, called CS-GNN, is then designed to improve the use of graph information based on the smoothness values of a graph. CS-GNN is shown to achieve better performance than existing methods in different types of real graphs.

## 1 Introduction

Graphs are powerful data structures that allow us to easily express various relationships (i.e., edges) between objects (i.e., nodes). In recent years, extensive studies have been conducted on GNNs for tasks such as node classification and link predication. GNNs utilize the relationship information in graph data and significant improvements over traditional methods have been achieved on benchmark datasets (Kipf & Welling, 2017; Hamilton et al., 2017; Velickovic et al., 2018; Xu et al., 2019; Hou et al., 2019). Such breakthrough results have led to the exploration of using GNNs and their variants in different areas such as computer vision (Satorras & Estrach, 2018; Marino et al., 2017), natural language processing (Peng et al., 2018; Yao et al., 2019), chemistry (Duvenaud et al., 2015), biology (Fout et al., 2017), and social networks (Wang et al., 2018). Thus, understanding why GNNs can outperform traditional methods that are designed for Euclidean data is important. Such understanding can help us analyze the performance of existing GNN models and develop new GNN models for different types of graphs.

In this paper, we make two main contributions: *(1) two graph smoothness metrics to help understand the use of graph information in GNNs*, and *(2) a new GNN model that improves the use of graph information using the smoothness values*. We elaborate the two contributions as follows.

One main reason why GNNs outperform existing Euclidean-based methods is because rich information from the neighborhood of an object can be captured. GNNs collect neighborhood information with aggregators (Zhou et al., 2018), such as the *mean* aggregator that takes the mean value of neighbors' feature vectors (Hamilton et al., 2017), the *sum* aggregator that applies summation (Duvenaud et al., 2015), and the *attention* aggregator that takes the weighted sum value (Velickovic et al., 2018). Then, the aggregated vector and a node's own feature vector are combined into a new feature vector. After some rounds, the feature vectors of nodes can be used for tasks such as node classification. Thus, the performance improvement brought by graph data is highly related to the *quantity* and *quality* of the neighborhood information. To this end, we propose *two smoothness metrics on node features and labels to measure the quantity and quality of neighborhood information of nodes*. The metrics are used to analyze the performance of existing GNNs on different types of graphs.

In practice, not all neighbors of a node contain relevant information w.r.t. a specific task. Thus, neighborhood provides both positive information and negative disturbance for a given task. Simply aggregating the feature vectors of neighbors with manually-picked aggregators (i.e., users choose

a type of aggregator for different graphs and tasks by trial or by experience) often cannot achieve optimal performance. To address this problem, we propose *a new model,* **CS-GNN**, *which uses the smoothness metrics to selectively aggregate neighborhood information to amplify useful information and reduce negative disturbance.* Our experiments validate the effectiveness of our two smoothness metrics and the performance improvements obtained by CS-GNN over existing methods.

## 2 MEASURING THE USEFULNESS OF NEIGHBORHOOD INFORMATION

We first introduce a general GNN framework and three representative GNN models, which show how existing GNNs aggregate neighborhood information. Then we propose two smoothness metrics to measure the quantity and quality of the information that nodes obtain from their neighbors.

### 2.1 GNN FRAMEWORK AND MODELS

The notations used in this paper, together with their descriptions, are listed in Appendix A. We use $\mathcal{G} = \{\mathcal{V}, \mathcal{E}\}$ to denote a graph, where $\mathcal{V}$ and $\mathcal{E}$ represent the set of nodes and edges of $\mathcal{G}$. We use $e_{v,v'} \in \mathcal{E}$ to denote the edge that connects nodes $v$ and $v'$, and $\mathcal{N}_v = \{v' : e_{v,v'} \in \mathcal{E}\}$ to denote the set of neighbors of a node $v \in \mathcal{V}$. Each node $v \in \mathcal{V}$ has a feature vector $x_v \in \mathcal{X}$ with dimension $d$. Consider a node classification task, for each node $v \in \mathcal{V}$ with a class label $y_v$, the goal is to learn a representation vector $h_v$ and a mapping function $f(\cdot)$ to predict the class label $y_v$ of node $v$, i.e., $\hat{y}_v = f(h_v)$ where $\hat{y}_v$ is the predicted label.

Table 1: Neighborhood aggregation schemes

| Models | Aggregation and combination functions for round $k$ ($1 \leq k \leq K$) |
|---|---|
| General GNN framework | $h_v^{(k)} = \text{COMBINE}^{(k)}\left(\left\{h_v^{(k-1)}, \text{AGGREGATE}^{(k)}\left(\{h_{v'}^{(k-1)} : v' \in \mathcal{N}_v\}\right)\right\}\right)$ |
| GCN | $h_v^{(k)} = A\left(\sum_{v' \in \mathcal{N}_v \cup \{v\}} \frac{1}{\sqrt{(|\mathcal{N}_v|+1) \cdot (|\mathcal{N}_{v'}|+1)}} \cdot W^{(k-1)} \cdot h_{v'}^{(k-1)}\right)$ |
| GraphSAGE | $h_v^{(k)} = A\left(W^{(k-1)} \cdot \left[h_v^{(k-1)} \| \text{AGGREGATE}(\{h_{v'}^{(k-1)}, v' \in \mathcal{N}_v\})\right]\right)$ |
| GAT | $h_{v_i}^{(k)} = A\left(\sum_{v_j \in \mathcal{N}_{v_i} \cup \{v_i\}} a_{i,j}^{(k-1)} \cdot W^{(k-1)} \cdot h_{v_j}^{(k-1)}\right)$ |

GNNs are inspired by the Weisfeiler-Lehman test (Weisfeiler & Lehman, 1968; Shervashidze et al., 2011), which is an effective method for graph isomorphism. Similarly, GNNs utilize a neighborhood aggregation scheme to learn a representation vector $h_v$ for each node $v$, and then use neural networks to learn a mapping function $f(\cdot)$. Formally, consider the general GNN framework (Hamilton et al., 2017; Zhou et al., 2018; Xu et al., 2019) in Table 1 with $K$ rounds of neighbor aggregation. In each round, only the features of 1-hop neighbors are aggregated, and the framework consists of two functions, AGGREGATE and COMBINE. We initialize $h_v^{(0)} = x_v$. After $K$ rounds of aggregation, each node $v \in \mathcal{V}$ obtains its representation vector $h_v^{(K)}$. We use $h_v^{(K)}$ and a mapping function $f(\cdot)$, e.g., a fully connected layer, to obtain the final results for a specific task such as node classification.

Many GNN models have been proposed. We introduce three representative ones: Graph Convolutional Networks (GCN) (Kipf & Welling, 2017), GraphSAGE (Hamilton et al., 2017), and Graph Attention Networks (GAT) (Velickovic et al., 2018). GCN merges the combination and aggregation functions, as shown in Table 1, where $A(\cdot)$ represents the activation function and $W$ is a learnable parameter matrix. Different from GCN, GraphSAGE uses concatenation '$\|$' as the combination function, which can better preserve a node's own information. Different aggregators (e.g., *mean*, *max pooling*) are provided in GraphSAGE. However, GraphSAGE requires users to choose an aggregator to use for different graphs and tasks, which may lead to sub-optimal performance. GAT addresses this problem by an attention mechanism that learns coefficients of neighbors for aggregation. With the learned coefficients $a_{i,j}^{(k-1)}$ on all the edges (including self-loops), GAT aggregates neighbors with a *weighted sum* aggregator. The attention mechanism can learn coefficients of neighbors in different graphs and achieves significant improvements over prior GNN models.

## 2.2 GRAPH SMOOTHNESS METRICS

GNNs usually contain an aggregation step to collect neighboring information and a combination step that merges this information with node features. We consider the *context* $c_v$ of node $v$ as the node's own information, which is initialized as the feature vector $x_v$ of $v$. We use $s_v$ to denote the *surrounding* of $v$, which represents the aggregated feature vector computed from $v$'s neighbors. Since the neighborhood aggregation can be seen as a convolution operation on a graph (Defferrard et al., 2016), we generalize the aggregator as weight linear combination, which can be used to express most existing aggregators. Then, we can re-formulate the general GNN framework as *a context-surrounding framework* with two mapping functions $f_1(\cdot)$ and $f_2(\cdot)$ in round $k$ as:

$$c_{v_i}^{(k)} = f_1(c_{v_i}^{(k-1)}, s_{v_i}^{(k-1)}), \quad s_{v_i}^{(k-1)} = f_2(\sum_{v_j \in \mathcal{N}_{v_i}} a_{i,j}^{(k-1)} \cdot c_{v_j}^{(k-1)}). \tag{1}$$

From equation (1), the key difference between GNNs and traditional neural-network-based methods for Euclidean data is that GNNs can integrate extra information from the surrounding of a node into its context. In graph signal processing (Ortega et al., 2018), features on nodes are regarded as signals and it is common to assume that observations contain both noises and true signals in a standard signal processing problem (Rabiner & Gold, 1975). Thus, we can decompose a context vector into two parts as $c_{v_i}^{(k)} = \check{c}_{v_i}^{(k)} + \breve{n}_{v_i}^{(k)}$, where $\check{c}_{v_i}^{(k)}$ is the true signal and $\breve{n}_{v_i}^{(k)}$ is the noise.

**Theorem 1.** *Assume that the noise $\breve{n}_{v_i}^{(k)}$ follows the same distribution for all nodes. If the noise power of $\breve{n}_{v_i}^{(k)}$ is defined by its variance $\sigma^2$, then the noise power of the surrounding input $\sum_{v_j \in \mathcal{N}_{v_i}} a_{i,j}^{(k-1)} \cdot c_{v_j}^{(k-1)}$ is $\sum_{v_j \in \mathcal{N}_{v_i}} (a_{i,j}^{(k-1)})^2 \cdot \sigma^2$.*

The proof can be found in Appendix B. Theorem 1 shows that the surrounding input has less noise power than the context when a proper aggregator (i.e., coefficient $a_{i,j}^{(k-1)}$) is used. Specifically, the *mean* aggregator has the best denoising performance and the *pooling* aggregator (e.g., max-pooling) cannot reduce the noise power. For the *sum* aggregator, where all coefficients are equal to 1, the noise power of the surrounding input is larger than that of the context.

### 2.2.1 FEATURE SMOOTHNESS

We first analyze the information gain from the surrounding without considering the noise. In the extreme case when the context is the same as the surrounding input, the surrounding input contributes no extra information to the context. To quantify the information obtained from the surrounding, we present the following definition based on information theory.

**Definition 2** (Information Gain from Surrounding). *For normalized feature space $\mathcal{X}_k = [0, 1]^{d_k}$, if $\sum_{v_j \in \mathcal{N}_{v_i}} a_{i,j}^{(k)} = 1$, the feature space of $\sum_{v_j \in \mathcal{N}_{v_i}} a_{i,j}^{(k)} \cdot \check{c}_{v_j}^{(k)}$ is also in $\mathcal{X}_k = [0, 1]^{d_k}$. The probability density function (PDF) of $\check{c}_{v_j}^{(k)}$ over $\mathcal{X}_k$ is defined as $C^{(k)}$, which is the ground truth and can be estimated by nonparametric methods with a set of samples, where each sample point $\check{c}_{v_i}^{(k)}$ is sampled with probability $|\mathcal{N}_{v_i}|/2|\mathcal{E}|$. Correspondingly, the PDF of $\sum_{v_j \in \mathcal{N}_{v_i}} a_{i,j}^{(k)} \cdot \check{c}_{v_j}^{(k)}$ is $S^{(k)}$, which can be estimated with a set of samples $\{\sum_{v_j \in \mathcal{N}_{v_i}} a_{i,j}^{(k)} \cdot \check{c}_{v_j}^{(k)}\}$, where each point is sampled with probability $|\mathcal{N}_{v_i}|/2|\mathcal{E}|$. The information gain from the surrounding in round $k$ can be computed by Kullback–Leibler divergence (Kullback & Leibler, 1951) as*

$$D_{KL}(S^{(k)} || C^{(k)}) = \int_{\mathcal{X}_k} S^{(k)}(\boldsymbol{x}) \cdot \log \frac{S^{(k)}(\boldsymbol{x})}{C^{(k)}(\boldsymbol{x})} d\boldsymbol{x}.$$

The Kullback–Leibler divergence is a measure of information loss when the context distribution is used to approximate the surrounding distribution (Kurt, 2017). Thus, we can use the divergence to measure the information gain from the surrounding into the context of a node. When all the context vectors are equal to their surrounding inputs, the distribution of the context is totally the same with that of the surrounding. In this case, the divergence is equal to 0, which means that there is no extra information that the context can obtain from the surrounding. On the other hand, if the context and the surrounding of a node have different distributions, the divergence value is strictly positive. Note that in practice, the ground-truth distributions of the context and surrounding signals are unknown.

In addition, for learnable aggregators, e.g., the attention aggregator, the coefficients are unknown. Thus, we propose a metric $\lambda_f$ to estimate the divergence. Graph smoothness (Zhou & Schölkopf, 2004) is an effective measure of the signal frequency in graph signal processing (Rabiner & Gold, 1975). Inspired by that, we define the *feature smoothness* on a graph.

**Definition 3** (*Feature Smoothness*). *Consider the condition of the first round, where $c_v^{(0)} = x_v$, we define the feature smoothness $\lambda_f$ over normalized space $\mathcal{X} = [0, 1]^d$ as*

$$\lambda_f = \frac{\left|\left| \sum_{v \in \mathcal{V}} \left( \sum_{v' \in \mathcal{N}_v} (x_v - x_{v'}) \right)^2 \right|\right|_1}{|\mathcal{E}| \cdot d},$$

*where $||\cdot||_1$ is the Manhattan norm.*

According to Definition 3, a larger $\lambda_f$ indicates that the feature signal of a graph has *higher frequency*, meaning that the feature vectors $x_v$ and $x_{v'}$ are more likely dissimilar for two connected nodes $v$ and $v'$ in the graph. In other words, nodes with dissimilar features tend to be connected. Intuitively, for a graph whose feature sets have high frequency, the context of a node can obtain more information gain from its surrounding. This is because the PDFs (given in Definition 2) of the context and the surrounding have the same probability but fall in different places in space $\mathcal{X}$. Formally, we state the relation between $\lambda_f$ and the information gain from the surrounding in the following theorem. For simplicity, we let $\mathcal{X} = \mathcal{X}_0$, $d = d_0$, $C = C^{(0)}$ and $S = S^{(0)}$.

**Theorem 4.** *For a graph $\mathcal{G}$ with the set of features $\mathcal{X}$ in space $[0, 1]^d$ and using the mean aggregator, the information gain from the surrounding $D_{KL}(S||C)$ is positively correlated to its feature smoothness $\lambda_f$, i.e., $D_{KL}(S||C) \sim \lambda_f$. In particular, $D_{KL}(S||C) = 0$ when $\lambda_f = 0$.*

The proof can be found in Appendix C. According to Theorem 4, a large $\lambda_f$ means that a GNN model can obtain much information from graph data. Note that $D_{KL}(S||C)$ here is under the condition when using the *mean* aggregator. Others aggregators, e.g., *pooling* and *weight* could have different $D_{KL}(S||C)$ values, even if the feature smoothness $\lambda_f$ is a constant.

### 2.2.2 LABEL SMOOTHNESS

After quantifying the information gain with $\lambda_f$, we next study how to measure the effectiveness of information gain. Consider the node classification task, where each node $v \in \mathcal{V}$ has a label $y_v$, we define $v_i \simeq v_j$ if $y_{v_i} = y_{v_j}$. The surrounding input can be decomposed into two parts based on the node labels as

$$\sum_{v_j \in \mathcal{N}_{v_i}} a_{i,j}^{(k-1)} \breve{c}_{v_j}^{(k-1)} = \sum_{v_j \in \mathcal{N}_{v_i}} \mathbb{I}(v_i \simeq v_j) a_{i,j}^{(k-1)} \breve{c}_{v_j}^{(k-1)} + \sum_{v_j \in \mathcal{N}_{v_i}} (1 - \mathbb{I}(v_i \simeq v_j)) a_{i,j}^{(k-1)} \breve{c}_{v_j}^{(k-1)},$$

where $\mathbb{I}(\cdot)$ is an indicator function. The first term includes neighbors whose label $y_{v_j}$ is the same as $y_{v_i}$, and the second term represents neighbors that have different labels. Assume that the classifier has good linearity, the label of the surrounding input is shifted to $\sum_{v_j \in \mathcal{N}_{v_i}} a_{i,j}^{(k-1)} \cdot y_{v_j}$ (Zhang et al., 2018), where the label $y_{v_j}$ is represented as a one-hot vector here. Note that in GNNs, even if the context and surrounding of $v_i$ are combined, the label of $v_i$ is still $y_{v_i}$. Thus, for the node classification task, it is reasonable to consider that neighbors with the same label contribute positive information and other neighbors contribute negative disturbance.

**Definition 5** (*Label Smoothness*). *To measure the quality of surrounding information, we define the label smoothness as*

$$\lambda_l = \sum_{e_{v_i,v_j} \in \mathcal{E}} \left(1 - \mathbb{I}(v_i \simeq v_j)\right)/|\mathcal{E}|.$$

According to Definition 5, a larger $\lambda_l$ implies that nodes with different labels tend to be connected together, in which case the surrounding contributes more negative disturbance for the task. In other words, a small $\lambda_l$ means that a node can gain much positive information from its surrounding. To use $\lambda_l$ to qualify the surrounding information, we require labeled data for the training. When some graphs do not have many labeled nodes, we may use a subset of labeled data to estimate $\lambda_l$, which is often sufficient for obtaining good results as we show for the BGP dataset used in our experiments.

In summary, we propose a context-surrounding framework, and introduce two smoothness metrics to estimate *how much information that the surrounding can provide* (i.e., larger $\lambda_f$ means more information) and *how much information is useful* (i.e., smaller $\lambda_l$ means more positive information) for a given task on a given graph.

## 3 CONTEXT-SURROUNDING GRAPH NEURAL NETWORKS

In this section, we present a new GNN model, called **CS-GNN**, which utilizes the two smoothness metrics to improve the use of the information from the surrounding.

### 3.1 THE USE OF SMOOTHNESS FOR CONTEXT-SURROUNDING GNNS

The aggregator used in CS-GNN is *weighted sum* and the combination function is *concatenation*. To compute the coefficients for each of the $K$ rounds, we use a multiplicative attention mechanism similar to Vaswani et al. (2017). We obtain $2|\mathcal{E}|$ attention coefficients by multiplying the leveraged representation vector of each neighbor of a node with the node's context vector, and applying the softmax normalization. Formally, each coefficient $a_{i,j}^{(k)}$ in round $k$ is defined as follows:

$$a_{i,j}^{(k)} = \frac{\exp\left(A(p_{v_i}^{(k)} \cdot q_{i,j}^{(k)})\right)}{\sum_{v_l \in \mathcal{N}_{v_i}} \exp\left(A(p_{v_i}^{(k)} \cdot q_{i,l}^{(k)})\right)}, \tag{2}$$

where $p_{v_i}^{(k)} = (W_p^{(k)} \cdot h_{v_i}^{(k)})^\top$, $q_{i,j}^{(k)} = p_{v_i}^{(k)} - W_q^{(k)} \cdot h_{v_j}^{(k)}$, $W_p^{(k)}$ and $W_q^{(k)}$ are two learnable matrices.

To improve the use of the surrounding information, we utilize feature and label smoothness as follows. First, we use $\lambda_l$ to drop neighbors with negative information, i.e., we set $a_{i,j}^{(k)} = 0$ if $a_{i,j}^{(k)}$ is less than the value of the $r$-th ($r = \lceil 2|\mathcal{E}|\lambda_l \rceil$) smallest attention coefficient. As these neighbors contain noisy disturbance to the task, dropping them is helpful to retain a node's own features.

Second, as $\lambda_f$ is used to estimate the quantity of information gain, we use it to set the dimension of $p_{v_i}^{(k)}$ as $\lceil d_k \cdot \sqrt{\lambda_f} \rceil$, which is obtained empirically to achieve good performance. Setting the appropriate dimension is important because a large dimension causes the attention mechanism to fluctuate while a small one limits its expressive power.

Third, we compute the attention coefficients differently from GAT (Velickovic et al., 2018). GAT uses the leveraged representation vector $W^{(k)} \cdot h_{v_j}^{(k)}$ to compute the attention coefficients. In contrast, in equation (2) we use $q_{i,j}^{(k)}$, which is the difference of the context vector of node $v_i$ and the leveraged representation vector of neighbor $v_j$. The definition of $q_{i,j}^{(k)}$ is inspired by the fact that a larger $\lambda_f$ indicates that the features of a node and its neighbor are more dissimilar, meaning that the neighbor can contribute greater information gain. Thus, using $q_{i,j}^{(k)}$, we obtain a larger/smaller $a_{i,j}^{(k)}$ when the features of $v_i$ and its neighbor $v_j$ are more dissimilar/similar. For example, if the features of a node and its neighbors are very similar, then $q_{i,j}^{(k)}$ is small and hence $a_{i,j}^{(k)}$ is also small.

Using the attention coefficients, we perform $K$ rounds of aggregations with the *weighted sum* aggregator to obtain the representation vectors for each node as

$$h_{v_i}^{(k)} = A\left(W_l^{(k)} \cdot \left(h_{v_i}^{(k-1)} \| \sum_{v_j \in \mathcal{N}_{v_i}} a_{i,j}^{(k-1)} \cdot h_{v_j}^{(k-1)}\right)\right),$$

where $W_l^{(k)}$ is a learnable parameter matrix to leverage feature vectors. Then, for a task such as node classification, we use a fully connected layer to obtain the final results $\hat{y}_{v_i} = A(W \cdot h_{v_i}^{(K)})$, where $W$ is a learnable parameter matrix and $\hat{y}_{v_i}$ is the predicted classification result of node $v_i$.

### 3.2 SIDE INFORMATION ON GRAPHS

Real-world graphs often contain *side information* such as attributes on both nodes and edges, local topology features and edge direction. We show that CS-GNN can be easily extended to include rich side information to improve performance. Generally speaking, side information can be divided into two types: context and surrounding. Usually, the side information attached on nodes belongs to the context and that on edges or neighbors belongs to the surrounding. To incorporate the side information into our CS-GNN model, we use the local topology features as an example.

We use a method inspired by GraphWave (Donnat et al., 2018), which uses heat kernel in spectral graph wavelets to simulate heat diffusion characteristics as topology features. Specifically, we construct $|\mathcal{V}|$ subgraphs, $\mathbb{G} = \{G_{v_1}, G_{v_2}..., G_{v_{|\mathcal{V}|}}\}$, from a graph $\mathcal{G} = \{\mathcal{V}, \mathcal{E}\}$, where $G_{v_i}$ is composed

of $v$ and its neighbors within $K$ hops (usually $K$ is small, $K = 2$ as default in our algorithm), as well as the connecting edges. For each $G_{v_i} \in \mathbb{G}$, the local topology feature vector $t_{v_i}$ of node $v_i$ is obtained by a method similar to GraphWave.

Since the topology feature vector $t_{v_i}$ itself does not change during neighborhood aggregation, we do not merge it into the representation vector. In the attention mechanism, we regard $t_{v_i}$ as a part of the context information by incorporating it into $p_{v_i}^{(k)} = (W_p^{(k)} \cdot (h_{v_i}^{(k)} || t_{v_i}))^\top$. And in the last fully connected layer, we use $t_{v_i}$ to obtain the predicted class label $\hat{y}_{v_i} = A(W \cdot (h_{v_i}^{(K)} || t_{v_i}))$.

### 3.3 COMPARISON WITH EXISTING GNN MODELS

The combination functions of existing GNNs are given in Table 1. The difference between additive combination and concatenation is that concatenation can retain a node's own feature. For the aggregation functions, different from GCN and GraphSAGE, GAT and CS-GNN improve the performance of a task on a given graph by an attention mechanism to learn the coefficients. However, CS-GNN differs from GAT in the following ways. First, the attention mechanism of CS-GNN follows multiplicative attention, while GAT follows additive attention. Second, CS-GNN's attention mechanism utilizes feature smoothness and label smoothness to improve the use of neighborhood information as discussed in Section 3.1. This is unique in CS-GNN and leads to significant performance improvements over existing GNNs (including GAT) for processing some challenging graphs (to be reported in Section 4.2). Third, CS-GNN uses side information such as local topology features to further utilize graph structure as discussed in Section 3.2.

## 4 EXPERIMENTAL EVALUATION

We first compare CS-GNN with representative methods on the node classification task. Then we evaluate the effects of different feature smoothness and label smoothness on the performance of neural networks-based methods.

### 4.1 BASELINE METHODS, DATASETS, AND SETTINGS

**Baseline.** We selected three types of methods for comparison: *topology-based methods, feature-based methods, and GNN methods*. For each type, some representatives were chosen. The topology-based representatives are **struc2vec** (Ribeiro et al., 2017), **GraphWave** (Donnat et al., 2018) and **Label Propagation** (Zhu & Ghahramani, 2002), which only utilize graph structure. struc2vec learns latent representations for the structural identity of nodes by random walk (Perozzi et al., 2014), where node degree is used as topology features. GraphWave is a graph signal processing method (Ortega et al., 2018) that leverages heat wavelet diffusion patterns to represent each node and is capable of capturing complex topology features (e.g., loops and cliques). Label Propagation propagates labels from labeled nodes to unlabeled nodes. The feature-based methods are **Logistic Regression** and **Multilayer Perceptron** (**MLP**), which only use node features. The GNN representatives are **GCN**, **GraphSAGE** and **GAT**, which utilize both graph structure and node features.

**Datasets.** We used five real-world datasets: three citation networks (i.e., Citeseer, Cora (Sen et al., 2008) PubMed (Namata et al., 2012)), one computer co-purchasing network in Amazon (McAuley et al., 2015), and one Border Gateway Protocol (BGP) Network (Luckie et al., 2013). The BGP network describes the Internet's inter-domain structure and only about 16% of the nodes have labels. Thus, we created two datasets: BGP (full), which is the original graph, and BGP (small), which was obtained by removing all unlabeled nodes and edges connected to them. The details (e.g., statistics and descriptions) of the datasets are given in Appendix D.

**Settings.** We use F1-Micro score to measure the performance of each method for node classification. To avoid under-fitting, 70% nodes in each graph are used for training, 10% for validation and 20% for testing. For each baseline method, we set their the parameters either as their default values or the same as in CS-GNN. For the GNNs and MLP, the number of hidden layers (rounds) was set as $K = 2$ to avoid over-smoothing. More detailed settings are given in Appendix E.

Note that GraphSAGE allows users to choose an aggregator. We tested four aggregators for Graph-SAGE (details in Appendix F) and report the best result for each dataset in our experiments below.

## 4.2 Performance Results of Node Classification

**Smoothness.** Table 2 reports the two smoothness values of each dataset. Amazon has a much larger $\lambda_f$ value (i.e., $89.67 \times 10^{-2}$) than the rest, while PubMed has the smallest $\lambda_f$ value. This implies that the feature vectors of most nodes in Amazon are dissimilar and conversely for PubMed. For label smoothness $\lambda_l$, BGP (small) has a fairly larger value (i.e., $0.71$) than the other datasets, which means that $71\%$ of connected nodes have different labels. Since BGP (full) contains many unlabeled nodes, we used BGP (small)'s $\lambda_l$ as an estimation.

Table 2: Smoothness values

| Smoothness value \\ Dataset Metrics | Citeseer | Cora | PubMed | Amazon | BGP (small) | BGP (full) |
|---|---|---|---|---|---|---|
| Feature Smoothness $\lambda_f$ ($10^{-2}$) | 2.76 | 4.26 | 0.91 | 89.67 | 7.46 | 5.90 |
| Label Smoothness $\lambda_l$ | 0.26 | 0.19 | 0.25 | 0.22 | 0.71 | ≈0.71 |

**F1-Micro scores.** Table 3 reports the F1-Micro scores of the different methods for the task of node classification. The F1-Micro scores are further divided into three groups. For the topology-based methods, Label Propagation has relatively good performance for the citation networks and the co-purchasing Amazon network, which is explained as follows. Label Propagation is effective in community detection and these graphs contain many community structures, which can be inferred from their small $\lambda_l$ values. This is because a small $\lambda_l$ value means that many nodes have the same class label as their neighbors, while nodes that are connected together and in the same class tend to form a community. In contrast, for the BGP graph in which the role (class) of the nodes is mainly decided by topology features, struc2vec and GraphWave give better performance. GraphWave ran out of memory (512 GB) on the larger graphs as it is a spectrum-based method. For the feature-based methods, Logistic Regression and MLP have comparable performance on all the graphs.

For the GNN methods, GCN and GraphSAGE have comparable performance except on the PubMed and BGP graphs, and similar results are observed for GAT and CS-GNN. The main reason is that PubMed has a small $\lambda_f$, which means that a small amount of information gain is obtained from the surrounding, and BGP has large $\lambda_l$, meaning that most information obtained from the surrounding is negative disturbance. Under these two circumstances, using concatenation as the combination function allows GraphSAGE and CS-GNN to retain a node's own features. This is also why Logistic Regression and MLP also achieve good performance on PubMed and BGP because they only use the node features. However, for the other datasets, GAT and CS-GNN have considerably higher F1-Micro scores than all the other methods. Overall, CS-GNN is the only method that achieves competitive performance on all the datasets.

Table 3: Node classification results

| F1-Micro(%) \\ Dataset Alg. | Citeseer | Cora | PubMed | Amazon | BGP (small) | BGP (full) |
|---|---|---|---|---|---|---|
| struc2vec | 30.98 | 41.34 | 47.60 | 39.86 | 48.40 | 49.66 |
| GraphWave | 28.12 | 31.66 | OOM | 37.33 | 50.26 | OOM |
| Label Propagation | 71.07 | 86.26 | 78.52 | 88.90 | 34.05 | 36.82 |
| Logistic Regression | 69.96 | 76.62 | 87.97 | 85.89 | 65.34 | 62.41 |
| MLP | 70.51 | 73.40 | 87.94 | 86.46 | **67.08** | 67.00 |
| GCN | 71.27 | 80.92 | 80.31 | 91.17 | 51.26 | 54.46 |
| GraphSAGE | 69.47 | 83.61 | 87.57 | 90.78 | 65.29 | 64.67 |
| GAT | 74.69 | 90.68 | 81.65 | 91.75 | 47.44 | 58.87 |
| CS-GNN (w/o LTF) | 73.58 | 90.38 | 89.42 | 92.48 | 66.20 | **68.83** |
| CS-GNN | **75.71** | **91.26** | **89.53** | **92.77** | 66.39 | 68.76 |

We further examine the effects of local topology features (LTF) on the performance. We report the results of CS-GNN without using LTF, denoted as **CS-GNN (w/o LTF)** in Table 3. The results show that using LTF does not significantly improve the performance of CS-GNN. However, the results do reveal the effectiveness of the smoothness metrics in CS-GNN, because the difference between

CS-GNN (w/o LTF) and GAT is mainly in the use of the smoothness metrics in CS-GNN's attention mechanism. As shown in Table 3, CS-GNN (w/o LTF) still achieves significant improvements over GAT on PubMed (by improving the gain of positive information) and BGP (by reducing negative noisy information from the neighborhood).

**Improvements over non-GNN methods.** We also evaluate whether GNNs are always better methods, in other words, whether graph information is always useful. Table 4 presents the improvements of existing GNNs (i.e., GCN, GraphSAGE, GAT) and CS-GNN over the topology-based and feature-based methods, respectively. The improvements (in %) are calculated based on the average F1-Micro scores of each group of methods. The results show that using the topology alone, even if the surrounding neighborhood is considered, is not sufficient. This is true even for the BGP graphs for which the classes of nodes are mainly determined by the graph topology. Compared with feature-based methods, GNN methods gain more information from the surrounding, which is converted into performance improvements. However, for graphs with small $\lambda_f$ and large $\lambda_l$, existing GNN methods fail to obtain sufficient useful information or obtain too much negative noisy information, thus leading to even worse performance than purely feature-based methods. In contrast, CS-GNN utilizes smoothness to increase the gain of positive information and reduce negative noisy information for a given task, thus achieving good performance on all datasets.

Table 4: Improvements of GNNs over non-GNN methods

| Improvement (%)     Dataset
Alg. | Citeseer | Cora | PubMed | Amazon | BGP (small) | BGP (full) |
|---|---|---|---|---|---|---|
| Existing GNNs vs. topology-based methods | 65% | 60% | 32% | 65% | 23% | 37% |
| CS-GNN vs. topology-based methods | 74% | 72% | 42% | 68% | 50% | 59% |
| Existing GNNs vs. feature-based methods | 2% | 13% | -5% | 6% | -17% | -9% |
| CS-GNN vs. feature-based methods | 8% | 22% | 2% | 8% | 0% | 6 % |

## 4.3 SMOOTHNESS ANALYSIS

The results in Section 4.2 show that GNNs can achieve good performance by gaining surrounding information in graphs with large $\lambda_f$ and small $\lambda_l$. However, the experiments were conducted on different graphs and there could be other factors than just the smoothness values of the graphs. Thus, in this experiment we aim to verify the effects of smoothness using one graph only.

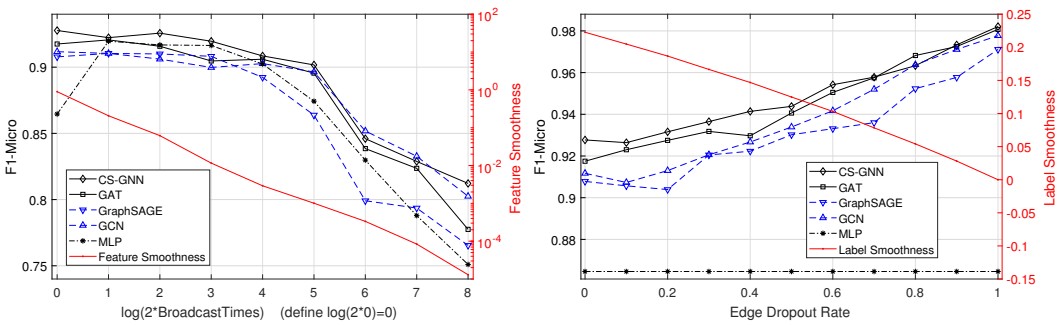

Figure 1: The effects of smoothness

To adjust $\lambda_f$ in a graph, we broadcast the feature vector of each node to its neighbors in rounds. In each round, when a node receives feature vectors, it updates its feature vector as the mean of its current feature vector and those feature vectors received, and then broadcast the new feature vector to its neighbors. If we keep broadcasting iteratively, all node features converge to the same value due to over-smoothness. To adjust $\lambda_l$, we randomly drop a fraction of edges that connect two nodes with different labels. The removal of such edges decreases the value of $\lambda_l$ and allows nodes to gain more positive information from their neighbors. We used the Amazon graph for the evaluation because the graph is dense and has large $\lambda_f$.

Figure 1 reports the F1-Micro scores of the neural-network-based methods (i.e., MLP and the GNNs). Figure 1 (left) shows that as we broadcast from $2^0$ to $2^8$ rounds, $\lambda_f$ also decreases accordingly. As $\lambda_f$ decreases, the performance of the GNN methods also worsens due to over-smoothness. However, the performance of MLP first improves significantly and then worsens, and becomes the poorest at the end. This is because the GNN methods can utilize the surrounding information by their design but MLP cannot. Thus, the broadcast of features makes it possible for MLP to first attain the surrounding information. But after many rounds of broadcast, the effect of over-smoothness becomes obvious and MLP's performance becomes poor. The results are also consistent with the fact that GNN models cannot be deep due to the over-smoothness effect. Figure 1 (right) shows that when $\lambda_l$ decreases, the performance of the GNN methods improves accordingly. On the contrary, since MLP does not use surrounding information, dropping edges has no effect on its performance. In summary, Figure 1 further verifies that GNNs can achieve good performance on graphs with large $\lambda_f$ and small $\lambda_l$, where they can obtain more positive information gain from the surrounding.

## 5 CONCLUSIONS

We studied how to measure the quantity and quality of the information that GNNs can obtain from graph data. We then proposed CS-GNN to apply the smoothness measures to improve the use of graph information. We validated the usefulness of our method for measuring the smoothness values of a graph for a given task and that CS-GNN is able to gain more useful information to achieve improved performance over existing methods.

### ACKNOWLEDGMENTS

We thank the reviewers for their valuable comments. We thank Mr. Ng Hong Wei for processing the BGP data. This work was supported in part by ITF 6904945 and GRF 14222816.

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

## A    NOTATIONS

The notations used in the paper and their descriptions are listed in Table 5.

Table 5: Notations and their descriptions

| Notations | Descriptions |
|---|---|
| $\mathcal{G}$ | A graph |
| $\mathcal{V}$ | The set of nodes in a graph |
| $v/v_i$ | Node $v/v_i$ in $\mathcal{V}$ |
| $\mathcal{E}$ | The set of edges in a graph |
| $e_{v_i,v_j}$ | An edge that connects nodes $v_i$ and $v_j$ |
| $\mathcal{X}$ | The node feature space |
| $x_v$ | The feature vector of node $v$ |
| $y_v/\hat{y}_v$ | The ground-truth / predicted class label of node $v$ |
| $\mathcal{N}_v$ | The set of neighbors of node $v$ |
| $h_v/h_v^{(k)}$ | The representation vector of node $v$ (in round $k$) |
| $f(\cdot)$ | A mapping function |
| $W$ | A parameter matrix |
| $A(\cdot)$ | An activation function |
| $a_{i,j}^{(k)}$ | The coefficient of node $v_j$ to node $v_i$ (in round $k$) |
| $c_v^{(k)}$ | The context vector of node $v$ (in round $k$) |
| $\check{c}_v^{(k)}$ | The ground-truth context vector of node $v$ (in round $k$) |
| $\check{n}_v^{(k)}$ | The noise on the context vector of node $v$ (in round $k$) |
| $s_v^{(k)}$ | The surrounding vector of node $v$ (in round $k$) |
| $d_k$ | The dimension of a representation vector (in round $k$) |
| $C^{(k)}$ | Probability density function (PDF) estimated by the term $\check{c}_v^{(k)}$ (in round $k$) |
| $S^{(k)}$ | Probability density function (PDF) estimated by the term $\sum_{v_j \in \mathcal{N}_{v_i}} a_{i,j}^{(k)} \cdot \check{c}_v^{(k)}$ (in round $k$) |
| $D_{KL}(S||C)$ | The Kullback–Leibler divergence between $S$ and $C$ |
| $\lambda_f$ | Feature smoothness |
| $\lambda_l$ | Label smoothness |
| $\|$ | Vector concatenation |
| $\|\cdot\|_1$ | Manhattan norm |
| $\mathbb{I}(\cdot)$ | An indicator function |
| $\boldsymbol{a}$ | Attention parameters (vector) |
| $t_v$ | Topology feature of node $v$ |
| $G_{v_i}$ | A subgraph built based on node $v_i$ |
| $\mathbb{G}$ | The set of subgraphs $G_{v_i}$ |

## B    PROOF OF THEOREM 1

*Proof.* We use the variance of $\check{n}_{v_j}^{(k-1)}$ to measure the power of noise. Without loss of generality, we assume that the signal $\check{c}_{v_j}^{(k-1)}$ is uncorrelated to the noise $\check{n}_{v_j}^{(k-1)}$ and noise is random, with a mean of zero and constant variance. If we define

$$\mathrm{Var}\big(\check{n}_{v_j}^{(k-1)}\big) = \mathbb{E}\big[\big(\check{n}_{v_j}^{(k-1)}\big)^2\big] = \sigma^2,$$

then after the weight aggregation, the noise power of $\sum_{v_j \in \mathcal{N}_{v_i}} a_{i,j}^{(k-1)} \cdot c_{v_j}^{(k-1)}$ is

$$\mathrm{Var}\Big(\sum\nolimits_{v_j \in \mathcal{N}_{v_i}} a_{i,j}^{(k-1)} \cdot \check{n}_{v_j}^{(k-1)}\Big) = \mathbb{E}\Big[\Big(\sum\nolimits_{v_j \in \mathcal{N}_{v_i}} a_{i,j}^{(k-1)} \cdot \check{n}_{v_j}^{(k-1)}\Big)^2\Big]$$

$$= \sum\nolimits_{v_j \in \mathcal{N}_{v_i}} \big(a_{i,j}^{(k-1)}\sigma\big)^2$$

$$= \sigma^2 \cdot \sum\nolimits_{v_j \in \mathcal{N}_{v_i}} \big(a_{i,j}^{(k-1)}\big)^2.$$

$\square$

## C   PROOF OF THEOREM 4

*Proof.* For simplicity of presentation, let $\mathcal{X} = \mathcal{X}_0$, $d = d_0$, $C = C^{(0)}$ and $S = S^{(0)}$. For $D_{KL}(S||C)$, since the PDFs of $C$ and $S$ are unknown, we use a nonparametric way, histogram, to estimate the PDFs of $C$ and $S$. Specifically, we uniformly divide the feature space $\mathcal{X} = [0,1]^d$ into $r^d$ bins $\{H_1, H_2, ..., H_{r^d}\}$, whose length is $\frac{1}{r}$ and dimension is $d$. To simplify the use of notations, we use $|H_i|_C$ and $|H_i|_S$ to denote the number of samples that are in bin $H_i$. Thus, we have

$$D_{KL}(S||C) \approx D_{KL}(\hat{S}||\hat{C})$$

$$= \sum_{i=1}^{r^d} \frac{|H_i|_S}{2|\mathcal{E}|} \cdot \log \frac{\frac{|H_i|_S}{2|\mathcal{E}|}}{\frac{|H_i|_C}{2|\mathcal{E}|}}$$

$$= \frac{1}{2|\mathcal{E}|} \cdot \sum_{i=1}^{r^d} |H_i|_S \cdot \log \frac{|H_i|_S}{|H_i|_C}$$

$$= \frac{1}{2|\mathcal{E}|} \cdot \left( \sum_{i=1}^{r^d} |H_i|_S \cdot \log |H_i|_S - \sum_{i=1}^{r^d} |H_i|_S \cdot \log |H_i|_C \right)$$

$$= \frac{1}{2|\mathcal{E}|} \cdot \left( \sum_{i=1}^{r^d} |H_i|_S \cdot \log |H_i|_S - \sum_{i=1}^{r^d} |H_i|_S \cdot \log \left( |H_i|_S + \Delta_i \right) \right),$$

where $\Delta_i = |H_i|_C - |H_i|_S$. Regard $\Delta_i$ as an independent variable, we consider the term $\sum_{i=1}^{r^d} |H_i|_S \cdot \log \left( |H_i|_S + \Delta_i \right)$ with second-order Taylor approximation at point 0 as

$$\sum_{i=1}^{r^d} |H_i|_S \cdot \log \left( |H_i|_S + \Delta_i \right) \approx \sum_{i=1}^{r^d} |H_i|_S \cdot \left( \log |H_i|_S + \frac{\ln 2}{|H_i|_S} \cdot \Delta_i - \frac{\ln 2}{2(|H_i|_S)^2} \cdot \Delta_i^2 \right).$$

Note that the numbers of samples for the context and the surrounding are the same, where we have

$$\sum_{i=1}^{r^d} |H_i|_C = \sum_{i=1}^{r^d} |H_i|_S = 2 \cdot |\mathcal{E}|.$$

Thus, we obtain

$$\sum_{i=1}^{r^d} \Delta_i = 0.$$

Therefore, the $D_{KL}(\hat{S}||\hat{C})$ can be written as

$$D_{KL}(S||C) \approx D_{KL}(\hat{S}||\hat{C})$$

$$= \frac{1}{2|\mathcal{E}|} \cdot \left( \sum_{i=1}^{r^d} |H_i|_S \cdot \log |H_i|_S - \sum_{i=1}^{r^d} |H_i|_S \cdot \log \left( |H_i|_S + \Delta_i \right) \right)$$

$$\approx \frac{1}{2|\mathcal{E}|} \left( \sum_{i=1}^{r^d} |H_i|_S \log |H_i|_S - \sum_{i=1}^{r^d} |H_i|_S \left( \log |H_i|_S + \frac{\ln 2}{|H_i|_S} \Delta_i - \frac{\ln 2}{2(|H_i|_S)^2} \Delta_i^2 \right) \right)$$

$$= \frac{1}{2|\mathcal{E}|} \cdot \sum_{i=1}^{r^d} \left( \frac{\ln 2}{2|H_i|_S} \Delta_i^2 - \ln 2 \cdot \Delta_i \right)$$

$$= \frac{\ln 2}{4|\mathcal{E}|} \cdot \sum_{i=1}^{r^d} \frac{\Delta_i^2}{|H_i|_S},$$

which is the Chi-Square distance between $|H_i|_C$ and $|H_i|_S$. If we regard $|H_i|_S$ as constant, then we have: if $\Delta_i^2$ are large, the information gain $D_{KL}(S||C)$ tends to be large. Formally, consider the

samples of $C$ as $\{x_v : v \in \mathcal{V}\}$ and the samples of $S$ as $\{\frac{1}{|\mathcal{N}_v|} \sum_{v' \in \mathcal{N}_v} x_{v'} : v \in \mathcal{V}\}$ with counts $|\mathcal{N}_v|$ of node $v$, we have the expectation and variance of their distance as

$$\mathbb{E}\left[|\mathcal{N}_v| \cdot x_v - \sum_{v' \in \mathcal{N}_v} x_{v'}\right] = 0,$$

$$\text{Var}\left(|\mathcal{N}_v| \cdot x_v - \sum_{v' \in \mathcal{N}_v} x_{v'}\right) = \mathbb{E}\left[\left(|\mathcal{N}_v| \cdot x_v - \sum_{v' \in \mathcal{N}_v} x_{v'}\right)^2\right] \geq 0.$$

For simplicity, for the distribution of the difference between the surrounding and the context, $|\mathcal{N}_v| \cdot x_v - \sum_{v' \in \mathcal{N}_v} x_{v'}$, we consider it as noises on the "expected" signal as the surrounding, $\frac{1}{|\mathcal{N}_v|} \cdot \sum_{v' \in \mathcal{N}_v} x_{v'}$, where the context $x_v$ is the "observed" signal. Apparently the power of the noise on the samples is positively correlated with the difference between their PDFs, which means that a large $\text{Var}\left(|\mathcal{N}_v| \cdot x_v - \sum_{v' \in \mathcal{N}_v} x_{v'}\right)$ would introduce a large difference between $|H_i|_S$ and $|H_i|_C$. Then, we obtain

$$\sum_{i=1}^{r^d} \frac{\Delta_i^2}{|H_i|_S} \sim \text{Var}\left(|\mathcal{N}_v| \cdot x_v - \sum_{v' \in \mathcal{N}_v} x_{v'}\right),$$

then it is easy to obtain

$$D_{KL}(S||C) \approx \frac{\ln 2}{4|\mathcal{E}|} \cdot \sum_{i=1}^{r^d} \frac{\Delta_i^2}{|H_i|_S} \sim \text{Var}\left(|\mathcal{N}_v| \cdot x_v - \sum_{v' \in \mathcal{N}_v} x_{v'}\right).$$

Recall the definition of $\lambda_f$, if $x_{v_i}$ and $x_{v_j}$ are independent, we have

$$\lambda_f = \frac{\left|\left|\sum_{v \in \mathcal{V}} \left(\sum_{v' \in \mathcal{N}_v}(x_v - x_{v'})\right)^2\right|\right|_1}{|\mathcal{E}| \cdot d}$$

$$= \frac{\left|\left|\sum_{v \in \mathcal{V}} \left(|\mathcal{N}_v| \cdot x_v - \sum_{v' \in \mathcal{N}_v} x_{v'}\right)^2\right|\right|_1}{|\mathcal{E}| \cdot d}$$

$$= \frac{\left|\left|\text{Var}\left(|\mathcal{N}_v| \cdot x_v - \sum_{v' \in \mathcal{N}_v} x_{v'}\right)\right|\right|_1}{|\mathcal{V}| \cdot |\mathcal{E}| \cdot d}$$

$$\sim \text{Var}\left(|\mathcal{N}_v| \cdot x_v - \sum_{v' \in \mathcal{N}_v} x_{v'}\right).$$

Therefore, we obtain

$$D_{KL}(S||C) \approx \frac{\ln 2}{4|\mathcal{E}|} \cdot \sum_{i=1}^{r^d} \frac{\Delta_i^2}{|H_i|_S} \sim \text{Var}\left(|\mathcal{N}_v| \cdot x_v - \sum_{v' \in \mathcal{N}_v} x_{v'}\right) \sim \lambda_f.$$

If $\lambda_f = 0$, we have that all nodes have the same feature vector $x_v$. Obviously, $D_{KL}(\hat{S}||\hat{C}) = 0$. □

## D  DATASETS

Table 6: Dataset statistics

| Dataset | Citeseer | Cora | PubMed | Amazon (computer) | BGP (small) | BGP (full) |
|---|---|---|---|---|---|---|
| $|\mathcal{V}|$ | 3,312 | 2,708 | 19,717 | 13,752 | 10,176 | 63,977 |
| $|\mathcal{E}|$ | 4,715 | 5,429 | 44,327 | 245,861 | 206,799 | 349,606 |
| Average degree | 1.42 | 2.00 | 2.25 | 17.88 | 20.32 | 5.46 |
| feature dim. | 3,703 | 1,433 | 500 | 767 | 287 | 287 |
| classes num. | 6 | 7 | 3 | 10 | 7 | 7 |
| $\lambda_f$ ($10^{-2}$) | 2.7593 | 4.2564 | 0.9078 | 89.6716 | 7.4620 | 5.8970 |
| $\lambda_l$ | 0.2554 | 0.1900 | 0.2455 | 0.2228 | 0.7131 | $\approx$0.7131 |

Table 6 presents some statistics of the datasets used in our experiments, including the number of nodes $|\mathcal{V}|$, the number of edges $|\mathcal{E}|$, the average degree, the dimension of feature vectors, the number

of classes, the feature smoothness $\lambda_f$ and the label smoothness $\lambda_l$. The BGP (small) dataset was obtained from the original BGP dataset, i.e., BGP (full) in Table 6, by removing all unlabeled nodes and edges connected to them. Since BGP (full) contains many unlabeled nodes, we used BGP (small)'s $\lambda_l$ as an estimation. As we can see in Table 6, the three citation networks are quite sparse (with small average degree), while the other two networks are denser. According to the feature smoothness, Amazon (computer) has much larger $\lambda_f$ than that of the others, which means that nodes with dissimilar features tend to be connected. As for the label smoothness $\lambda_l$, the BGP network has larger value than that of the others, meaning that connected nodes tend to belong to different classes. A description of each of these datasets is given as follows.

- Citeseer (Sen et al., 2008) is a citation network of Machine Learning papers that are divided into 6 classes: {Agents, AI, DB, IR, ML, HCI}. Nodes represent papers and edges model the citation relationships.

- Cora (Sen et al., 2008) is a citation network of Machine Learning papers that divided into 7 classes: {Case Based, Genetic Algorithms, Neural Networks, Probabilistic Methods, Reinforcement Learning, Rule Learning, Theory}. Nodes represent papers and edges model the citation relationships.

- PubMed (Namata et al., 2012) is a citation network from the PubMed database, which contains a set of articles (nodes) related to diabetes and the citation relationships (edges) among them. The node features are composed of TF/IDF-weighted word frequencies, and the node labels are the diabetes type addressed in the articles.

- Amazon Product (McAuley et al., 2015) is a co-purchasing network derived from the Amazon platform. Nodes are computers and edges connect products that were bought together. Features are calculated from the product image, and labels are the categories of computers. We used the processed version by Alex Shchur in GitHub.

- Border Gateway Protocol (BGP) Network (Luckie et al., 2013) describes the Internet's inter-domain structure, where nodes represent the autonomous systems and edges are the business relationships between nodes. The features contain basic properties, e.g., the location and topology information (e.g., transit degree), and labels means the types of autonomous systems. We used the dataset that was collected in 2018-12-31 and published in *Center for Applied Internet Data Analysis*.

## E  PARAMETERS

For struc2vec and GraphWave, we used their default settings to obtain the node embeddings for all nodes since they are unsupervised. For struc2vec, we set their three optimization as "True". The embedding dimension of struc2vec is 128 and that of GraphWave is 100. Then based on those embeddings, we put them in a logistic model implemented by (Perozzi et al., 2014), where the embeddings of the training set were used to train the logistic model and the embeddings of the test set were used to obtain the final F1-Micro score. Label Propagation is a semi-supervised algorithm, where nodes in the training set were regarded as nodes with labels, and those nodes in the validation set and test set were regarded as unlabeled nodes.

The feature-based methods and GNN methods used the same settings: the batch size was set as 512; the learning rate was set as 0.01; the optimizer was Adam. Except GraphSAGE, which set the epoch number as 10, all the other methods were implemented with the early stop strategy with the patience number set as 100. Other parameters were slightly different for different datasets but still the same for all methods. Specifically, for Citeseer and Cora, the dropout was set as 0.2, and the weight decay was set as 0.01. And the hidden number was set as 8. For PubMed, the dropout was set as 0.3, but the weight decay was set as 0. The hidden number was 16. For Amazon (computer), BGP (small) and BGP (full), the dropout was set as 0.3, and the weight decay was set as 0. The hidden number was 32. As for GAT and CS-GNN, the attention dropout was set as the same as the dropout. And the dimension of topology features in CS-GNN was set as 64. Note that the residual technique introduced in GAT was used for the GNN methods. As for the activation function in CS-GNN, *ReLU* was used for feature leveraging and *ELU* was used for the attention mechanism.

Table 7: The F1-Micro scores of GraphSAGE with different aggregators

| F1-Micro (%) Dataset / Aggregators | Citeseer | Cora | PubMed | Amazon | BGP (small) | BGP (full) |
|---|---|---|---|---|---|---|
| GCN | 71.27 | 80.92 | 80.31 | 91.17 | 51.26 | 54.46 |
| GraphSAGE-GCN | 68.57 | **83.61** | 81.76 | 88.14 | 49.64 | 48.54 |
| GraphSAGE-mean | 69.02 | 82.31 | 87.42 | **90.78** | 64.96 | 63.76 |
| GraphSAGE-LSTM | 69.17 | 82.50 | 87.08 | 90.09 | **65.29** | **64.67** |
| GraphSAGE-pool (max) | **69.47** | 82.87 | **87.57** | 87.39 | 65.06 | 64.24 |

## F    THE PERFORMANCE OF GRAPHSAGE WITH DIFFERENT AGGREGATORS

Table 7 reports the F1-Micro scores of GCN and GraphSAGE with four different aggregators: GCN, mean, LSTM and max-pooling. The results show that for GraphSAGE, except on PubMed and BGP, the four aggregators achieve comparable performance. The results of GCN and GraphSAGE-GCN are worse than the others for PubMed and the BGP graphs because of the small information gain (i.e., small $\lambda_f$) of PubMed and the large negative disturbance (i.e., large $\lambda_l$) of BGP as reported in Table 3 and Section 4.2. As explained in Section 3.3, GCN uses additive combination merged with aggregation, where the features of each node are aggregated with the features of its neighbors. As a result, GCN has poor performance for graphs with small $\lambda_f$ and large $\lambda_l$, because it merges the context with the surrounding with negative information for a given task. This is further verified by the results of GraphSAGE using the other three aggregators, which still have comparable performance on all the datasets. In Section 4, we report the best F1-Micro score among these four aggregators for each dataset as the performance of GraphSAGE.

## G    THE KULLBACK–LEIBLER DIVERGENCE VS. MUTUAL INFORMATION

We use the Kullback–Leibler Divergence (KLD), instead of using Mutual Information (MI), to measure the information gain from the neighboring nodes in Section 2.2.1 because of the following reason. In an information diagram, mutual information $I(X;Y)$ can be seen as the overlap of two correlated variables $X$ and $Y$, which is a symmetric measure. In contrast, $D_{KL}(X||Y)$ can be seen as the extra part brought by $X$ to $Y$, which is a measure of the non-symmetric difference between two probability distributions. Considering the node classification task, the information contributed by neighbors and the information contributed to neighbors are different (i.e., non-symmetric). Thus, we use the KLD instead of MI.

We remark that, although some existing GNN works (Velickovic et al., 2019; Chen et al., 2019) use MI in their models, their purposes are different from our work. MI can be written as $I(X,Y) = D_{KL}(P(X,Y)||P(X) \times P(Y))$, where $P(X,Y)$ is the joint distribution of $X$ and $Y$, and $P(X)$, $P(Y)$ are marginal distributions of $X$ and $Y$. From this perspective, we can explain the mutual information of $X$ and $Y$ as the information loss when the joint distribution is used to approximate the marginal distributions. However, this is not our purpose in node classification.

