# OpenReview forum: "Measuring and Improving the Use of Graph Information in Graph Neural Networks"
_ICLR.cc/2020/Conference — Accept (Poster)_

### Official Review · AnonReviewer3 · 2019-10-21
**Official Blind Review #3**

**Rating:** 8

**Review:**

The authors study how neighbor information on graphs can be used in Graph Neural Networks. It proposes measures on whether the data in neighboring nodes are useful in terms of labels or features. It also provides a new Graph Neural Network algorithm that is a modification of attention-based models incorporating the derived label and feature smoothness measures. The paper demonstrates the usefulness of these measures and algorithms with several different baselines from different families. The writing is mostly smooth, and the authors seem to provide enough detail of the experiments performed.

The proposed measures look simple but effective (as demonstrated in Table 2). The paper compares different techniques and shows that when neighboring labels are not smooth, techniques such as label propagation does not help. I do recommend Table 4 Lambda f and Lambda l values to be included in the main paper (though text mentions). When incorporated into the attention-based GNNs, Figure 1 also shows that smoothness parameters of the techniques also improve.

**Experience Assessment:**

I have read many papers in this area.

**Review Assessment: Checking Correctness Of Derivations And Theory:**

I assessed the sensibility of the derivations and theory.

**Review Assessment: Checking Correctness Of Experiments:**

I assessed the sensibility of the experiments.

**Review Assessment: Thoroughness In Paper Reading:**

I read the paper at least twice and used my best judgement in assessing the paper.

---

> ### Author Response · Authors · 2019-11-09
> **Reporting Lambda f and Lambda l values in the main paper**
>
> Thank you for your positive feedbacks and the suggestion. We have now included a separate table (Table 2) in the main paper to report and discuss Lambda f and Lambda l values of the datasets. Indeed, this should make the analysis in Section 4.2 easier to follow as the two smoothness metrics are one of our key contributions to measure and understand the use of graph information in GNNs.

---

### Official Review · AnonReviewer2 · 2019-10-25
**Official Blind Review #2**

**Rating:** 3

**Review:**

This paper proposes two smoothness metrics to measure the quantity and quality of the graph information that GNNs employ in the model. With the smoothness measures, this paper proposes a new GNN model to improve the use of graph information. Overall, the paper is well-organized and clearly written. The main concern is the novelty, since the proposed method is pretty close to the graph attention network (GAT), except using the two smoothness metrics and a slightly different way to compute the attention coefficients. Experimental results show marginal improvement over existing GNN methods on the node classification task. Given these aspects, it is not that convincing that the introduced smoothness metrics are necessary. I would like to recommend a weak reject for this paper.


Suggestions to improve the paper:

1) It would be better to provide more convincing evidence and motivation for the smoothness metrics, either in theory or empirical analysis.

2) When describing the proposed method, organize the key differences in a more clear way. For example, add the proposed method directly in Table 1 and summarize the key differences in bullet points. Currently, this important part is deferred to Section 3.3, which may cause confusion for the readers to understand the paper.

3) In Section 3.2, this paper claims that the proposed method can easily include side information on graphs to improve performance, by using the topology feature in the last fully connected layer for class prediction. However, this technique can also be used in existing GNN models, and it is not clear why this is described as something unique for the proposed method. Also, there is no corresponding ablation experiments to compare the performance of the proposed method with and without using local topology features.

**Experience Assessment:**

I have published one or two papers in this area.

**Review Assessment: Checking Correctness Of Derivations And Theory:**

I assessed the sensibility of the derivations and theory.

**Review Assessment: Checking Correctness Of Experiments:**

I assessed the sensibility of the experiments.

**Review Assessment: Thoroughness In Paper Reading:**

I read the paper at least twice and used my best judgement in assessing the paper.

---

> ### Author Response · Authors · 2019-11-09
> **[Part 1] Response to the reviewer's concern on the similarity with GAT**
>
> Thank you for your comments and the detailed suggestions. We try to address your main concerns below and hopefully this will make the contributions of our paper clearer.
>
> First, we would like to discuss your concern that our method is close to GAT.  While our GNN model, i.e., CS-GNN, is also an attention-based method like GAT, we remark that our contributions are not just a new GNN model itself. In addition to CS-GNN, another main contribution of our work is the use of the two smoothness metrics to help researchers understand what and how much graph information can benefit an existing GNN model (not just our model). For example, we found that existing GNNs, e.g., GAT, achieve good performance for graphs with large feature smoothness and small label smoothness. However, they do not perform well for graphs with small feature smoothness or large label smoothness because they fail to obtain sufficient useful information or obtain too much negative information from neighboring nodes.
>
> Such understanding is very important as until now there is limited knowledge about why and in what situations GNNs can outperform other methods. For example, we show in Table 3 that GAT outperforms MLP in some cases, but GAT has significantly worse results than MLP for processing graphs with small feature smoothness or large label smoothness. Also, label propagation can have similar or even better performance than GNN methods such as GCN and GraphSAGE for processing some graphs.
>
> In our paper, we used the two smoothness measures to explain the performance of different methods on different graph datasets. And we believe the insights obtained in our study are valuable to researchers, as they can also use the smoothness measures to better understand the performance of different GNNs on different graphs (e.g., designing better models, building benchmarks). As currently there is a lack of metrics to measure the quantity and quality of information a GNN obtains from graph data, our smoothness metrics may also inspire other researchers to develop better measures to gain more comprehensive understanding.
>
> Our second contribution is the CS-GNN model, which improves the use of graph information using the smoothness values. Although CS-GNN is also an attention-based method, there are some key differences between CS-GNN and GAT as we will explain in our response to Suggestion 2 below. In addition, CS-GNN actually achieves quite significant performance improvements over GAT as we will explain in our next response.
>
> We admit that we did not make our contributions clear in the paper, and this has obscured the primary purpose of the smoothness metrics. We have now stated in the 2nd paragraph of Section 1 the two main contributions of the paper. We hope the reviewer would find this clearer now. Thank you for raising this concern.

---

> ### Author Response · Authors · 2019-11-09
> **[Part 2] Response to the reviewer's concern on the marginal performance improvements**
>
> Next, we would like to address your concern that our experimental results show marginal improvement over existing GNN methods. Actually we believe the improvements made by CS-GNN is quite significant, but maybe the numbers are buried in the table as we squeezed many things in Table 2, which has now been divided into 3 tables. CS-GNN outperforms GAT by 9.65% and 17.14% on the PubMed and BGP graphs. CS-GNN outperforms GraphSAGE by 8.98%, 9.15% and 6.32% on the Citeseer, Cora and BGP graphs. CS-GNN outperforms GCN by 6.23%, 12.78%, 11.45% and 26.26% on the Citeseer, Cora, PubMed and BGP graphs. Although in some other cases the performance may seem to be marginal, there is still improvement in every case and we would like to remark that such marginal improvements are actually considered “acceptable” in related work. For example, Velickovic et al. (2018) reported that GAT improves over GCN only for 2.26%, 1.97% and 0.00% on Citeseer, Cora and PubMed. But this is understandable as it is very challenging for a method to outperform other methods by a large margin in every case. Thus, while some of our improvements may be marginal, we would like to highlight that there are actually quite a number of other cases that CS-GNN does obtain significant improvements over existing methods. We understand these numbers could have been buried in the big table in the previous version, and we hope that by breaking the table into three smaller tables, the results are clearer now.

---

> ### Author Response · Authors · 2019-11-09
> **[Part 3] Responses to the reviewer's three suggestions**
>
> Response to Suggestion 1: Thank you for your suggestion. We believe one of the main problems is that we did not clearly highlight our two main contributions, and thus the first contribution, i.e., the smoothness metrics are proposed primarily to understand how the use of graph information may (or may not) benefit GNNs (as detailed in our response to the first concern above), was not clear to readers. To address your concern, we have revised our paper to clearly state the purposes of proposing the smoothness metrics, which should have strengthened the motivation for the smoothness metrics.
>
> In addition, we have also added another comparison result, CS-GNN (no topology), so that the difference between CS-GNN (no topology) and GAT is mainly in the use of the smoothness metrics in CS-GNN’s attention mechanism. The results reported in Table 3 show that CS-GNN (no topology) still achieves significant improvements over GAT on the PubMed and BGP graphs. The improvements are mainly because the use of smoothness enables CS-GNN to increase the gain of positive information and reduce negative noisy information from these two graphs.
>
>
>
> Response to Suggestion 2: Thank you for your suggestion. We did not include CS-GNN in Table 1 because Section 2.1 only introduces the background and we do not find it easy to include CS-GNN there without introducing some details of our method first, and thus this may make readers even more confused. Instead, we now list the key differences between CS-GNN and GAT clearly in Section 3.3 of the revised paper. First, the attention mechanism of CS-GNN is similar to the multiplicative attention mechanism, while GAT is similar to the additive attention mechanism. Compared with the traditional multiplicative attention mechanism, we introduced two weight matrices instead of one to compute the attention coefficients. Second, we consider the smoothness metrics in our attention mechanism, which is unique in CS-GNN and leads to significant performance improvements over existing GNNs (e.g., GAT) for processing graphs such as PubMed and BGP. Third, we also include the use of nodes’ local topology features in computing the attention coefficients to further improve the performance. (Remark: if the reviewer does not think the last point is significant enough, we may consider to remove it. Please also see our response to Suggestion 3 below.)
>
>
>
> Response to Suggestion 3: We have conducted experiments on CS-GNN without using local topology features (LTF), denoted as CS-GNN (w/o LTF). The results are presented in Table 3 in the revised paper, which shows that using LTF does improve the performance of CS-GNN, but not significantly. However, the results do reveal the effectiveness of the smoothness metrics in CS-GNN, because the difference between CS-GNN (w/o LTF) and GAT is mainly in the use of the smoothness metrics in CS-GNN’s attention mechanism. As shown in Table 3, CS-GNN (w/o LTF) still achieves significant improvements over GAT on the PubMed and BGP graphs.
>
> We also found that the use of LTF in other GNNs also only leads to marginal performance improvement, e.g., using LTF in GCN even has worse performance in a few cases. If the reviewer does not feel the improvements obtained using LTF very significant, we would consider not to put the use of side information as a subsection in Section 3.2, but only briefly discuss it at the end of Section 3.1. Thank you for suggesting the ablation experiments that helped us see the effects of LTF!

---

### Official Review · AnonReviewer1 · 2019-10-31
**Official Blind Review #1**

**Rating:** 8

**Review:**

Summary

The paper proposes two graph smoothness metrics for measuring the usefulness of graph information. The feature smoothness indicates how much information can be gained by aggregating neighboring nodes while the label smoothness assesses the quality of this information. The authors show that Graph Neural Networks (GNNs) work best for tasks with high features smoothness and low label smoothness by utilizing information from surrounding nodes which also tends to have the same label. Based on these two metrics, the authors introduce a framework, called Context-Surrounding Graph Neural Network (CS-GNN), that utilizes important information from neighboring nodes of the same label while reduce the disturbance from neighboring nodes from different classes. The results demonstrate considerable improvement across 5 different tasks.

Strength

The authors advocate for better understanding of the use of graph information in learning, which is both an important and interesting problem. Two graph smoothness metrics appears to be intuitive and reflect common situations in graph-based data (1) features from neighboring nodes contribute differently to target node representation (2) neighboring nodes information sometimes causing disturbance if node with different labels tend to be connected. The paper provides some theoretical analysis that supports this claim and thorough experiments that show the correlation between the two proposed metrics with the performance of GNNs.

Weakness

While the paper is reasonably readable, there is certainly room for improvements in the clarity of the paper. First, I would suggest the authors to avoid too much word repetition as well as long, obscure sentences. For example, the first sentence of section 2.2 can  be rewritten as “GNNs usually contains an aggregation step to collect neighboring information and a combination step that merges this information with node features.” The flow of the paper is also hard to follow and need some rearrangement. For instance, paragraph 3 of section 2.1 can be pushed until section 3.3. Another suggestion about the flow is to separate section 2.2 into two subsections for features smoothness and label smoothness (also the title of this section need to be refined). Finally, the results would be more clear if separated into different tables or subsections/paragraphs.

Questions

* Is there a particular reason for using the KLD instead of mutual information?
* In 2nd sentence of section 2.2, what does “node’s own information” mean? If it is the individual node’s features then why it naturally the representation vector h_v (which is aggregated with neighboring nodes?)

**Experience Assessment:**

I have read many papers in this area.

**Review Assessment: Checking Correctness Of Derivations And Theory:**

I assessed the sensibility of the derivations and theory.

**Review Assessment: Checking Correctness Of Experiments:**

I carefully checked the experiments.

**Review Assessment: Thoroughness In Paper Reading:**

I read the paper thoroughly.

---

> ### Author Response · Authors · 2019-11-09
> **Revisions according to the reviewer's suggestions**
>
> Thank you for your positive feedbacks and all your suggestions. We have made the following changes to our paper according to your suggestions:
>
> 1. Rephrased all long sentences in the paper.
>
> 2. Re-organized some subsections and tables. We separated Section 2.2 into two subsections and changed the title of Section 2.2. But we didn’t move paragraph 3 of Section 2.1 to Section 3.3 because paragraph 3 only discusses the background of existing GNNs listed in Table 1. Instead, we rewrote Section 3.3 to mainly discuss the key differences between existing GNNs and CS-GNN.
>
> 3. Added a new section in Appendix G to discuss the differences between the KLD and mutual information (also see our answer to your question below).
>
>
> Answers to other questions are given as follows.
>
> Q1: Is there a particular reason for using the KLD instead of mutual information?
>
> A1: We use the KLD instead of MI because of the following reason. In an information diagram, mutual information I(X; Y) can be seen as the overlap of two correlated variables X and Y, which is a symmetric measure. In contrast, KLD(X||Y) can be seen as the extra part brought by X to Y, which is a measure of the non-symmetric difference between two probability distributions. Considering the node classification task, the information contributed by neighbors and the information contributed to neighbors are different (i.e., non-symmetric). Thus, we use the KLD instead of MI.
>
> We remark that, although some existing GNN works [1, 2] use MI in their models, their purposes are different from our work. MI can be written as I(X, Y) = D_{KL}(P(X,Y)||P(X) \times P(Y)), where P(X,Y) is the joint distribution of X and Y, and P(X), P(Y) are marginal distributions of X and Y. From this perspective, we can explain the mutual information of X and Y as the information loss when the joint distribution is used to approximate the marginal distributions. However, this is not our purpose in node classification.
>
> We included the above discussion in Appendix G of the revised paper as some readers may be interested to know.
>
> Q2: In 2nd sentence of section 2.2, what does “node’s own information” mean? If it is the individual node’s features then why it naturally the representation vector h_v (which is aggregated with neighboring nodes?)
>
> A2: The “node’s own information” here refers to the node’s information before the aggregation with neighboring nodes takes place. That is, here h_v = h_v^{(0)} = x_v, i.e., the initial value of h_v or the feature vector x_v of v. To avoid confusion, we have modified the sentence as: We consider the context c_v of a node v as the node’s own information, which is initialized as the feature vector x_v of v.
>
> Thank you again for your suggestions, which have helped make the paper flow clearer and highlight our contributions.
>
> [1] Petar Velickovic, William Fedus, William L. Hamilton, Pietro Li{\`{o}}, Yoshua Bengio and R. Devon Hjelm. Deep Graph Infomax. In ICLR 2019.
>
> [2] Anonymous Authors. Utilizing Edge Features in Graph Neural Networks via Variational Information Maximization. Submitted to ICLR 2020.

---

### Decision · Program_Chairs · 2019-12-19

**Decision:**

Accept (Poster)

**Comment:**

Two reviewers are positive about this paper while the other reviewer is negative. The low-scoring reviewer did not respond to discussions. I also read the paper and found it interesting. Thus an accept is recommended.